# An Overview of Genus *Malachra* L.—Ethnobotany, Phytochemistry, and Pharmacological Activity

**DOI:** 10.3390/plants11212808

**Published:** 2022-10-22

**Authors:** Leonor Cervantes-Ceballos, Jorge Sánchez-Hoyos, Fredys Sanchez-Hoyos, Erick Torres-Niño, Jairo Mercado-Camargo, Amparo Echeverry-Gómez, Karick Jotty Arroyo, Esther del Olmo-Fernández, Harold Gómez-Estrada

**Affiliations:** 1Grupo de Investigación en Química Orgánica Medicinal, Facultad de Ciencias Farmacéuticas, Campus de Zaragocilla, University of Cartagena, Cartagena 130001, Colombia; 2Departamento de Ciencias Farmacéuticas, Área de Química Farmacéutica, Centro de Enfermedades Tropicales de la Universidad de Salamanca (CIETUS), Instituto de Investigación Biomédica de Salamanca (IBSAL), Facultad de Farmacia, Campus Miguel de Unamuno, Universidad de Salamanca, 37007 Salamanca, Spain

**Keywords:** *Malachra*, ethnobotany, pharmacological activity

## Abstract

The genus *Malachra* L. belongs to the family Malvaceae. It includes herbs or subshrubs of nine accepted species with approximately thirty synonyms, and it has been widely used in community folk medicine to treat health problems including inflammation, nasal obstruction, leishmaniasis, malaria, childbirth, kidney disorders, fever, respiratory tract diseases, among others. From the genus *Malachra* L., flavonoids, steroids, triterpenes, anthocyanins, leucoanthocyanins, saponins, carbohydrates, phenols, glycosides, and alkaloids have been isolated and identified. Some pharmacological reports have indicated that the genus has antidiarrheal, antiepileptic, antiulcerogenic, antioxidant, anticonvulsant, antiviral, anticancer, antibacterial, anthelmintic, and hepatoprotective properties. However, there have been limited studies of bioactive molecules with pharmacological and biological activities associated with *Malachra alceifolia* Jacq., *Malachra capitata* (L.) L., *Malachra fasciata* Jacq., *Malachra radiata* (L.) L., *Malachra ruderalis* Gürke., *Malachra rudis* Benth., *Malachra helodes* Mart., *Malachra urens* Poit. ex Ledeb. & Alderstam., and *Malachra officinalis* Klotzsch. In this review, we consider the conservation of these species to save the ancestral knowledge of their traditional use in populations, and their pharmacological potential for future studies in search of alternatives for solutions to diseases in humans and animals and tools for the design and search of potential bioactive compounds against infectious and non-infectious agents.

## 1. Introduction

Various populations of some regions of the world use medicinal plants to cure and prevent their health problems [1]. However, the World Health Organization (WHO) has defined ancestral and traditional knowledge resources obtained from plants, extracts, fractions, or compounds as a primary source for the identification of molecules, substances, or active principles to be used in the development of drugs and phytomedicines for disease prevention [2,3,4]. Secondary metabolites derived from plant species such as *Taxus brevifolia* (Paclitaxel) [5], *Catharanthus roseus* (Vincristine) [6], *Digitalis lanata* (Digoxin) [7], *Ephedra distachya* (Ephedrine) [8], *Artemisia annua* (Artemisinin) [9], *Salix alba* (acetylsalicylic acid) [10], and *Papaver somniferum* (morphine) [11] have been used as drugs with anticancer, cytotoxic, antimicrobial, analgesic, inflammatory, antiestrogenic, antiallergic, antioxidant, and other activities [12,13]. Note that ancestral and traditional community activities have used medicinal plants for phytotherapy in primary health care and for validation of their ethnopharmacological use [14,15].

There are about 382,000 accepted plant species names in the world [16,17]. The criteria for considering the acceptance, name, and synonyms of a plant species are defined by taxonomic concepts from botanical studies, herbarium specimens, biogeographical, conservation and phylogenomic research, as well as expert observations [18]. About 20% of these names are unresolved, indicating that some included data sources that did not provide any evidence or view on whether or not the name should be treated as accepted, or there were contradictory opinions that could not be easily resolved, leaving ambiguous and unchecked concepts [17].

*Malachra* L. is an accepted genus that belongs to the Malvaceae family and contains seventeen known species, with only nine accepted species including *Malachra alceifolia* Jacq., *Malachra capitata* (L.) L., *Malachra fasciata* Jacq., *Malachra radiata* (L.) L., *Malachra ruderalis* Gürke., *Malachra rudis* Benth., *Malachra helodes* Mart., *Malachra urens* Poit. ex Ledeb. & Alderstam [16], and *Malachra officinalis* Klotzsch [17]. However, only eight species of the genus *Malachra* L. are ambiguous and unchecked [7] wih some reports of botanical descriptions, ethnobotanicals, phytoconstituents, pharmacological, and biological activities. Therefore, some species names of the genus *Malachra* L. are currently unchecked, ambiguous, and awaiting taxonomic scrutiny: *Malachra lobata* L., *Malachra texana* A. Gray., *Malachra triloba* [Desf.], *Malachra rosea* Hoffmanns., *Malachra digitata* C. Presl., *Malachra urticifolia* C. Presl., *Malachra viminea* Fisch. ex Steud., and *Malachra plumosa* Desr. [8]; to date, there are no reported descriptions such as those mentioned above [18,19].

The life form of the *Malachra* L. genus is a herb or subshrub present in terrestrial substrate [19]. Regarding geographical distribution, the genus *Malachra* L. genus is native to approximately fifty countries in tropical and subtropical America and West Africa, and has been introduced in some regions of the United States, India, and Asia [20]. Colombia and Brazil are biodiverse countries that harbor most of the Earth’s species of the genus *Malachra* L. and high numbers of endemic species including *Malachra alceifolia* Jacq., *Malachra capitata* (L.) L., *Malachra fasciata* Jacq., *Malachra radiata* (L.) L., *Malachra ruderalis* Gürke., and *Malachra rudis* Benth. are distributed in almost all the territories. In Colombia, there are few studies on the conservation status according to the categories and criteria of the International Union for Conservation of Nature and Natural Resources (IUCN) and biological and pharmacological potentials [21].

The genus *Malachra* L. contains phytoconstituents isolated from whole plant, leaf, stem, flowers, and root, fresh or dried, flavonoids [22]; compounds of phenolic acids [23,24]; phytosterol [24,25]; fatty acid derivative [26]; alkaloids, tannin [27,28]; and dipeptide [29,30]. The phytoconstituents are molecules necessary for plant survival, but also contribute to the prevention of many infectious and non-infectious diseases. Several communities in these countries use these traditional plants for treating ailments such as inflammation, clogged nose [1], leishmaniasis [31], malaria [32,33], kidney disorder, fever [34], diseases of the respiratory tract [35], and during childbirth, and therefore, have become one of the best medicinal alternatives for these communities.

The species in the genus *Malachra* L. are frequently reported as weeds, in pastures and rangelands, and usually serving as food for livestock [36]. In addition to this, they fulfill functions of vital importance to the ecosystem, and they are a source of food for various species of butterflies, as well as a great repository for the larvae of the same [37]. At an ecosystemic level, the species in this genus are considered to be primarily for the regeneration processes of intervened ecosystems since they favor the succession processes of other species in these ecosystems [38].

In this review, we organize the ethnobotanical, phytochemical, biological, pharmacological, and toxicological information of plants belonging to the genus *Malachra* L., to analyze the state-of-the-art of the genus and to highlight the pharmacognostic potential of molecules extracted from these plants, organic extracts, and fractions as potential sources of new drugs and for the development of innovative medicines.

## 2. Material and Methods

This review was conducted using the Preferred Reporting Items for Systematic Review (PRISMA) guidelines for systematic reviews. First, in accordance with the PICO method of the PRISMA guidelines for data selection, a scientific question was posed with the following heading: P: genus *Malachra* L. in the family Malvaceae, I: Included species, C: accepted, ambiguous, and unchecked species, O: taxonomy, botanical descriptions, geographical distribution, phytochemistry, ethnobotany, pharmacological, and biological activities of species. Followed by the search strategy: ((*Malachra* [Title/Abstract])) OR, Search: (Taxonomy [MeSH Terms]) AND (genus *Malachra* [Title/Abstract])) AND (Taxonomy [Title/Abstract]) OR (*Malachra* species [Title/Abstract])) AND (*Malachra* species, ethnobotanical [Title/Abstract])), OR (*Malachra* species [Title/Abstract]) AND (*Malachra* species, phytoconstituents [Title/Abstract]) AND (*Malachra* species, ethnobotanical [Title/Abstract]) AND (*Malachra* species, pharmacological activity [Title/Abstract]) AND (*Malachra* species, biological activity [Title/Abstract]); the type of search: new; databases: Medline, Latin America & Iberia; Platforms: PubMed accessed on 10 January 2022 (https://pubmed.ncbi.nlm.nih.gov/, accessed on 1 March 2022), ScienceDirect accessed on 10 January 2022 (www.sciencedirect.com, accessed on 1 March 2022), Scopus accessed on 10 January 2022 (www.scopus.com, accessed on 1 March 2022), and Scielo accessed on 10 January 2022 (www.scielo.org, accessed on 1 March 2022); search date range: unlimited; Language Restrictions: none. Different inclusion and exclusion criteria were considered. The inclusion criteria used were as follows: Websites where the names of the species mentioned in this review have validated geographic distribution, and are recognized as taxonomic and botanical World Plants (www.worldplants.de), including World Flora Online (WFO) (http://www.worldfloraonline.org/, accessed on 1 March 2022), World Checklist of Vascular Plants (WCVP) (https://wcvp.science.kew.org/, accessed on 1 March 2022), Plants of the World Online (https://powo.science.kew.org/, accessed on 1 March 2022), Catálogo de plantas y líquenes de Colombia (http://catalogoplantasdecolombia.unal.edu.co/, accessed on 1 March 2022), Tropicos (https://www.tropicos.org/home, accessed on 1 March 2022), Tramil (https://www.tramil.net/es, accessed on 1 March 2022), Angiosperm Phylogeny (http://www.mobot.org/MOBOT/research/APweb/, accessed on 1 March 2022), and Global Biodiversity Information Facility (https://www.gbif.org/, accessed on 1 March 2022).Articles, abstracts, and university repositories with descriptions of phytoconstituents using clear methodology for obtaining extracts, fractions, compounds, and plant part used.Articles, abstracts, and university repositories with pharmacological and biological descriptions using clear methodology in the evaluation of doses, concentrations, biological models used, and controls.

The exclusion criteria were those studies that did not meet criteria one, two, and three.

There were fifteen bibliographic registers identified in the database; records deleted eight (congress abstract six, taxonomic website version not updated one (http://www.theplantlist.org/, accessed on 1 March 2022), article without pharmacological description one); articles evaluated for eligibility eight; articles accepted for the study seventy-nine; flora microsites accepted nine; graduate work four. 

## 3. Taxonomy the Genus *Malachra*

According to World Flora Online, the genus *Malachra* L. belongs to the family Malvaceae, the group of angiosperms with the largest accepted name reports, including 14,539 scientific species names, 4465 accepted species names, and 245 genera [39]. The botanical characteristics of this family describe its growth as shrub, tree, herb, liana/vine, subshrub, aquatic, terrestrial, hemiepiphytic, and rupicolous [17,40]. The genus *Malachra* L. has nine species with accepted names and prolific taxonomic synonyms with adjusted morphological descriptions [17]. However, in the Flora World website *Malachra officinalis* Klotzsch., is accepted, whereas in other flora websites such as https://powo.science.kew.org/, https://wcvp.science.kew.org/, and https://www.tramil.net/es, (accessed on 1 March 2022), this species is not accepted (Table 1 and Table 2). There is no information on this species provided in this review due to its limited studies.

## 4. Botanical Description of the Species of the Genus *Malachra* L.

The genus *Malachra* L. is represented by herbs or suffrutes, sometimes puberulent, generally hispid, or with stinging trichomes. Leaves are simple or palmately lobed; acute, acuminate, or obtuse at the apex; truncated or subcordate at the base; serrated or crenated; and generally pubescent. They also present: bracteate inflorescences; axillary or terminal heads; broadly cordate-ovate bracts at the base, with prominent nerves often alternating with whitish areas, although it can be in other parts green and sessile. Absent calliculus (except in *M*. *radiata*); small calyx, five-lobed; white, yellow, or lilac corolla; and five-lobed leaves can also be observed. Additionally, schizocarpal fruits have the shape of a wheel similar to a cheese that can be split into five ripe wing types; flowers are in heads, surrounded by bracts larger than them [17,41,42,43]. 

***Malachra alceifolia* Jacq.** is a herb or shrub, 1. 6 m tall. The aerial parts and petioles are hispid, simple or split, flavescent, tuberculate base, scattered hairs, short, star-shaped, and long forming knots, with stem pubescence and a green to reddish color. The leaves are simple, opposite, and lobed, with up to five lobes, with pubescence, and their margins are serrated or toothed. There are inflorescences in axillary heads with the presence of acuminate bracts and flowers with five petals, 2 to 3 cm in diameter, with a yellow coloration. The fruits have bracts and persistent calyx, with numerous seeds, up to 2 mm in diameter, dark colored, and flattened (Figure 1) [44].

***Malachra capitata*****Linn.** is a herb or subshrub, up to 2 m tall, with stem pubescence and green to orange coloration. The leaves are simple, opposite, palmate or lobed, with up to five lobes; the texture of the leaves is velvety and their margins are crenulate or serrated. Inflorescences are peduncular and axillary, with the presence of lanceolate bracts; flowers have five petals, 1 cm in diameter, with a white coloration. Schizocarpic fruits have single seeds up to 3 mm in diameter, and are dark colored (Figure 2) [45].

***Malachra fasciata* Jacq.** is a herb or subshrub, up to 2 m tall, with stem pubescence, simple or stellate trichomes up to 7 mm, and green to orange coloration. The leaves are simple, opposite, ovate, truncated at the base, acute at the apex, lobed or deeply parted, otherwise crenate-serrate, pubescent above with appressed trichomes. The heads are short-stalked bracts, 2–6 per head, lance/ovate, subcordate at the base, acute at the apex, often ciliate, prominently hispid. Calyx 4 to 5 mm, hispid; petals white 6–8 mm, mericarps light brown to grayish green, reticulate veined (Figure 3) [46].

***Malachra radiata* (L.) L.** is a herb or suffrutex, 1.5 m tall, with stem pubescence and green and purple coloration. The leaves are simple, opposite, palmate or lobed, with up to seven lobes; the texture of the leaves is velvety, and their margins are crenulate or serrated, and most apical leaves may have a triangular shape. Inflorescences are pedicular and terminal, with the presence of ovate or acute bracts; flowers have five petals, 2.5 cm in diameter, with a lilac coloration and purple at the base. Schizocarpic fruits have numerous seeds, up to 2 mm in diameter, and are dark colored (Figure 4) [47].

***Malachra ruderalis* Gürke.** is an annual herb, up to 3 m high, with stem pubescence and green coloration. The leaves are simple, opposite, palmate or lobed, with up to five lobes; the texture of the leaves is velvety, and their margins are crenulate; the most apical leaves have a triangular shape. It shows pedicular inflorescence with terminal or acute axillary bracts, five petals, 3 cm long, with yellow schizocarpic fruit with numerous dark-colored seeds up to 2 mm in diameter (Figure 5) [48,49].

***Malachra rudis* Benth**. is a perennial herb, up to 0.7 m high, with a stem that has very short pubescence, and its coloration is light green. The leaves are simple, opposite, lobed, with up to three lobes; velvety texture; crenulated margins with pedicular, axillary, or terminal inflorescence; ovate bracts with five petals; 1.5 cm in long. Schizocarpic fruits are yellow to white with numerous seeds up to 1 mm in diameter, one per carpel (Figure 6) [48,49].

***Malachra urens* Poit. ex Ledeb. & Alderstam** is an annual herb, up to 1 m high, with pubescent stems of different sizes and green to orange coloration. The leaves are simple, opposite and lobed, with up to three lobes relatively marked, with pubescence; the margins of the leaves are toothed, about 3–12 cm long. Inflorescences are peduncular racemes with acuminate, boat-shaped, pubescent bracts. Flowers have five petals, 2 to 3 cm in long, with a yellow color. Schizocarpic fruits have mericarps and pubescent bracts, with numerous seeds, up to 2 mm in long, dark colored, and flat (Figure 7) [48,49].

***Malachra helodes* Mart.** is an annual herb, up to 1 m high. The stem is slightly pubescent with dark to light green coloration. The leaves are simple, opposite, and lobed, with up to five lobes that are well marked, with slightly marked pubescence on the underside; the margins of the leaves are serrated, about 3–10 cm long. Inflorescences are terminal and axillary racemes with the presence of acuminate bracts in the form of pubescent. The flowers have five petals, 2 to 3 cm in length, with a pink color. Schizocarpic fruits with mericarps and bracts with stem indumentum hispid; terminal inflorescence; pinkish flower; epicalyx absent; fruit, up to 4 mm in length; glabrous indumentum (Figure 8) [48,49].

## 5. Geographical Distribution of the Genus *Malachra* L.

The genus *Malachra* L. is geographically distributed in tropical and subtropical America, west tropical Africa, and Southwest Asia [49]. 

The species of the genus *Malachra* L. are native to: South America countries (northeast Argentina, Bolivia, north Brazil, northeast Brazil, southeast Brazil, Colombia, Ecuador, French Guiana, Guyana, Paraguay, Peru, Suriname, and Venezuela); Central America countries (Belize, Costa Rica, El Salvador, Guatemala, Honduras, Nicaragua, and Panama); southwest Caribbean and Bahamas (Aruba, Cuba, Dominican Republic, Haiti, Jamaica, and Trinidad-Tobago); North America countries (Mexico); American Caribbean (Puerto Rico); Africa countries (Benin, Burkina, Congo, Chad, Ivory Coast, Mali, Ghana, Sudan, Niger, Nigeria, Senegal, and Togo); Caribbean Sea Islands (Netherlands Antilles, Leeward Is., Windward Is., and Venezuelan Antilles) [49]. Moreover, the genus has been introduced in: Asian countries (India, Thailand, Bangladesh, Taiwan, Philippine, and Indonesian); United States (Hawaii, Louisiana Florida, and Texas); Caribbean Sea Islands (Guadalupe, Martinique, and Granada) [49] (Figure 9).

The distribution of the species in Brazil and Colombia covers most of the regions [43,50]. The species with the highest abundance are found in Colombian Caribbean, denoting their preference for warmer climates. In terms of their ecology, they prefer open habitats such as the savannah and scrublands; however, they can occur less frequently in dry and humid forests. There have been a few reports on the conservation status of the species in the genus *Malchara* L. In this study, in Table 4, only the conservation status of *Malachra alceifolia* Jacq., in Colombia, is included. 

## 6. Species of the Genus *Malachra* L. with Ethnobotanical Use

Medicinal plants of traditional use have been included in the Traditional and Complementary Medicine (TCM) report by the World Health Organization (WHO). Many of them are used in developing countries such as Chile and Colombia, with usages of 71% and 40% of the population, respectively [51,52]. Table 5 lists and describes the species of the genus *Malachra* L. with reported medicinal use, including vernacular name, part of the plant used, condition treated, preparation and administration, and the country where its use has been reported. The species in the genus *Malachra* L. with the highest reported ethnobotanical use are *M. alceifolia* Jacq., (for treatment of inflammation, clogged nose, during malaria, childbirth, drunk headache, kidney disorder, fever, and headache), *M. capitata* (L.) L., (for pain, diarrhea, convulsion, hepatic cirrhosis, inflammation, pyrexia, ulcer, dementia, treatment of wounds, gastric disorders, jaundice, childbirth, malaria headache, and fever), *M. fasciata* Jacq., (for emollients, hemorrhoids, fever, impotence, gonorrhea, rheumatism, demulcent, and diuretic), *M. ruderalis* Gürke., (for pulmonary diseases, fever, cough, sore throat, flu, colds, whooping cough, pulmonary diseases, COVID-19, stomach problems, diarrhea, fever, skin spots, skin infections, gastritis, stress inflammation, and vaginal infection).

## 7. Phytochemistry Species of the Genus *Malachra* L.

Phytoconstituents are chemical compounds that plants synthesize as a defense mechanism against biotic and abiotic environmental conditions; they play key roles in biological processes [58,59]. Many phytoconstituents isolated from the genus *Malachra* L., such as flavonoids, coumarins, carbohydrates, glycosides, triterpenes, alkaloids, tannins, and saponins, exhibit pharmacologic activity. Table 6 lists the phytoconstituents reported only in *Malachra* species: *M. alceifolia* Jacq., *M. capitata* (L.) L., and *M. fasciata* Jacq, see Figure 10 molecular structure of some compounds isolated from species of the genus *Malachra* L.

Generally, organic compounds obtained from extracts, fractions, and isolated compounds of plant species possess unique structural characteristics among which the geometrical and energetic interactions of the atoms stand out; chirality and stereoisomers of molecular structures, intermolecular interactions, hydrogen bond acceptors and donors, molecular mass, diversity of ring systems, among others, become tools of interest for medicinal chemistry in drug development, to improve the potency, for pharmacokinetic properties, and to reduce the toxicity of new drugs [60]. However, the secondary metabolites isolated from plant species constitute a group of chemical molecules with a great diversity of biological activities applied to the pharmaceutical, cosmetic, and food sectors [61].

Gallic acid (GA) is a phenolic compound with anti-inflammatory, antimicrobial, hepatoprotective, neuroprotective, and carcinogenic properties that prevent gastrointestinal, cardiovascular, metabolic, and neuropsychological diseases [62]. The livers of Wistar rats exposed to carbon tetrachloride in doses of 50 mg/kg and 100 mg/kg GA, were evaluated by decreasing serum liver enzymes, regulating the expression of proinflammatory genes, and regulating the expression of antioxidant genes [63]; mercuric chloride induced at 200 mg/kg, increased glutathione peroxidase, superoxide dismutase, and catalase activity, and decreased the level of glutathione in liver tissue [64]. Caffeic acid has antimicrobial potential against *Staphylococcus aureus* strains with MICs from 256 µg/mL to 1024 µg/mL [65]. The compound quercetin isolated from extracts and organic fractions of *Allium cepa* L., *Morus alba*, *Camellia sinensis*, *Moringa oleifera*, and *Centella asiatica,* at doses between 50 and 100 mg/kg, and evaluated in in vivo models, showed antiulcer activity [66]. Beta-sitosterol isolated chloroform extract of *Corchorus capsularis L.* leaves has been shown to exhibit a significant effect against trypanothione reductase *Leishmania donovani* promastigotes at IC_50_  =  17.7  ±  0.43 µg/mL [67], and the dipeptide aurantiamide acetate patent has an effect of resisting influenza virus and an inhibition effect on a cytopathic effect mediated by the influenza A virus CN106431960B, filing date: 11 November 2018, legal status: active [68].

## 8. Species of the Genus *Malachra* L. with Pharmacological Activity

Applications of plant remedies in traditional medicine are still central in the health systems in some countries of world [69]. The biogenesis and biosynthesis of phytoconstituents in plant species provide an opportunity for medicinal chemistry to advance pharmacological studies for treating pathologies that have been little studied [70]. Table 7 summarizes the most important pharmacological activities reported for the genus *Malachra* L., such as antidiarrheal, anti-epileptic, antiulcerogenic, antioxidant, anticonvulsant, hepatoprotective, antiviral, anticancer, antibacterial, and anthelmintic properties. The bibliographic search describes only the pharmacological activities of *M. alceifolia* Jacq., *M. capitata* (L.) L., and *M. fasciata* Jacq. 

## 9. Conclusions

This review provides information and analysis of published scientific data on eight species of the genus *Malachra* L. of the family Malvaceae: *Malachra alceifolia* Jacq., *Malachra capitata* (L.) L., *Malachra fasciata* Jacq., *Malachra radiata* (L.) L., *Malachra ruderalis* Gürke., *Malachra rudis* Benth., *Malachra helodes* Mart., *Malachra urens* Poit., and *Malachra urens* Poit. ex Ledeb. & Alderstam. The plants are widely distributed in America, Africa, and Asia, with a greater distribution of species in the tropical and subtropical regions of Colombia and Brazil. These plant species have shown important applications in traditional medicine since their aqueous extracts are used for treating infectious, inflammatory, respiratory, digestive, and neurological problems. The active phytoconstituents isolated from the genus may be useful for the evaluation and identification of molecular targets against infectious pathogens or inflammatory processes. The presence of groups of metabolites such as flavonoids, sterol-terpenoids, and phenolics, including gallic acid, caffeic acid, quercetin, and β-sitosterol compounds may be associated with biological processes with activities such as antidiarrheal, antiepileptic, antiulcerogenic, antioxidant, anticonvulsant, hepatoprotective, antiviral, anticancer, antibacterial, and anthelmintic, with inflammatory activity being the most widespread. In turn, the presence of peptides with pharmacological potential, contribute to the search for drugs against oncological, metabolic, cardiovascular, and neglected tropical diseases. Therefore, this review contributes to the baseline knowledge for the search of information on the validation of the therapeutic use and conservation of traditional and ancestral knowledge of plant biodiversity in America, Africa, and Asia countries, as well as contributes to basic concepts for future research aimed at the discovery of new drugs.

## Figures and Tables

**Figure 1 plants-11-02808-f001:**
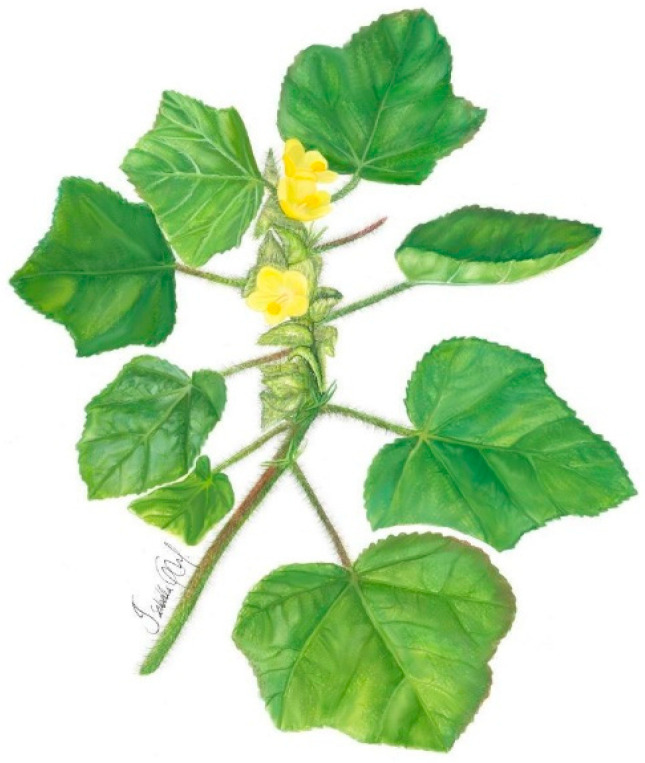
*Malachra alceifolia* Jacq.

**Figure 2 plants-11-02808-f002:**
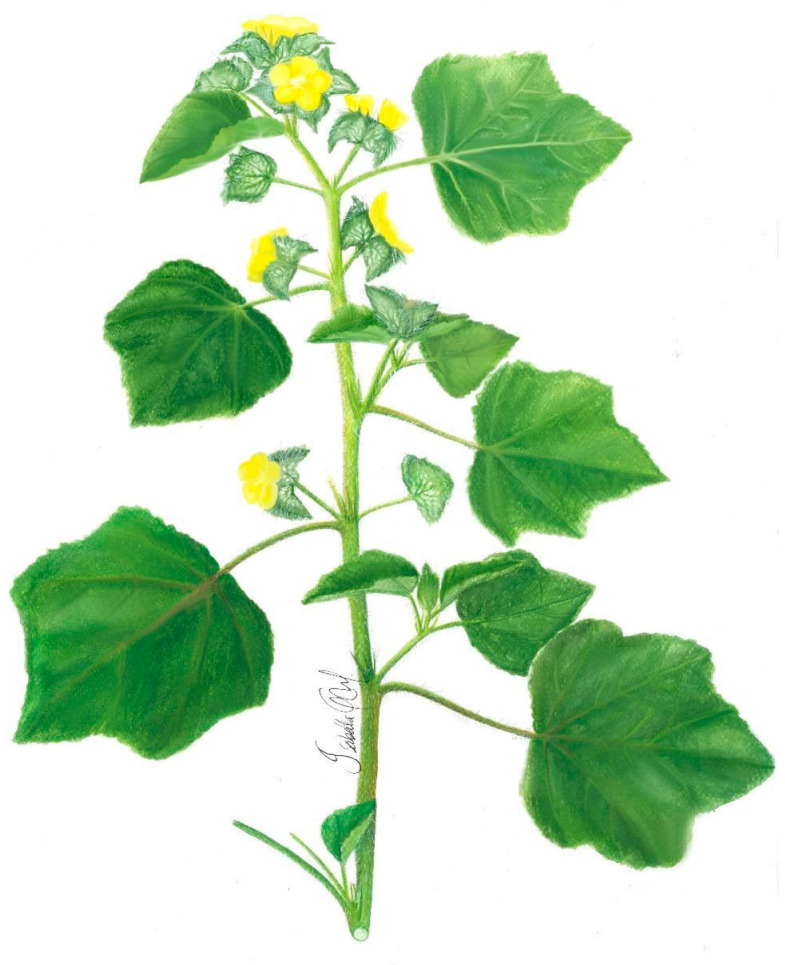
*Malachra capitata* Linn.

**Figure 3 plants-11-02808-f003:**
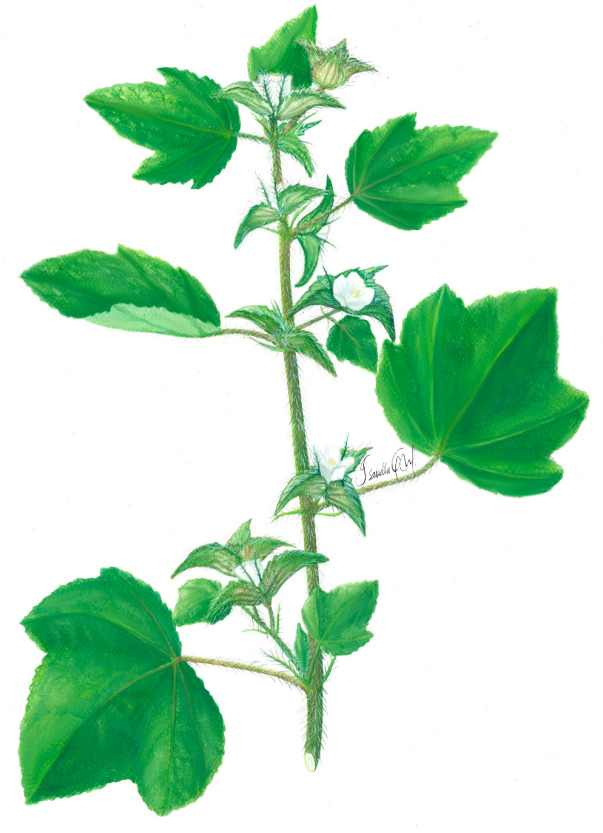
*Malachra fasciata* Jacq.

**Figure 4 plants-11-02808-f004:**
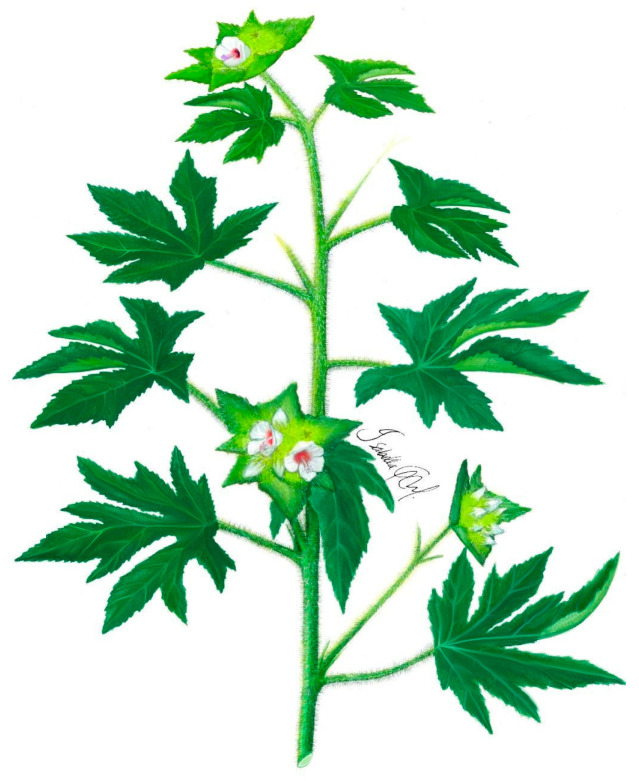
*Malachra radiata* (L.) L.

**Figure 5 plants-11-02808-f005:**
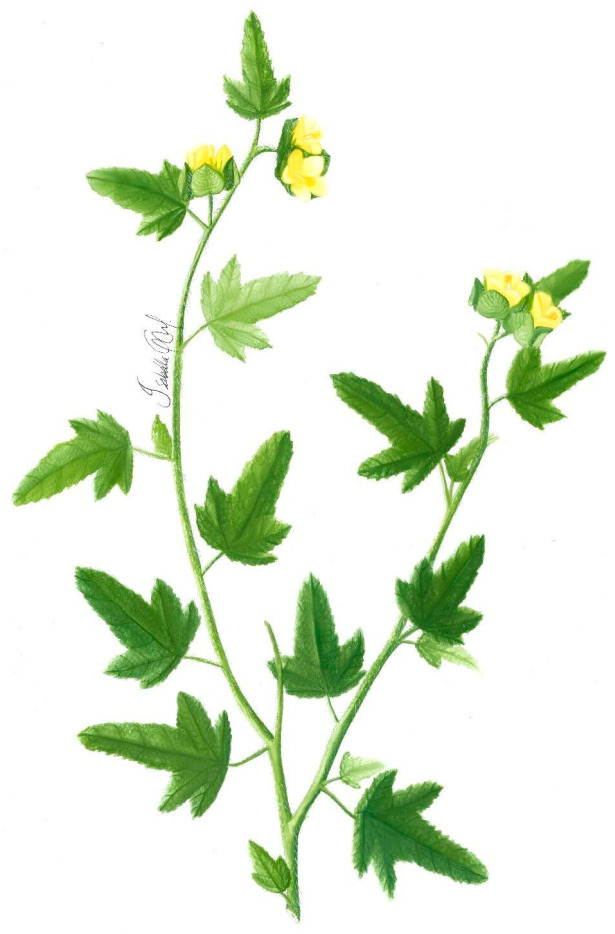
*Malachra ruderalis* Gürke.

**Figure 6 plants-11-02808-f006:**
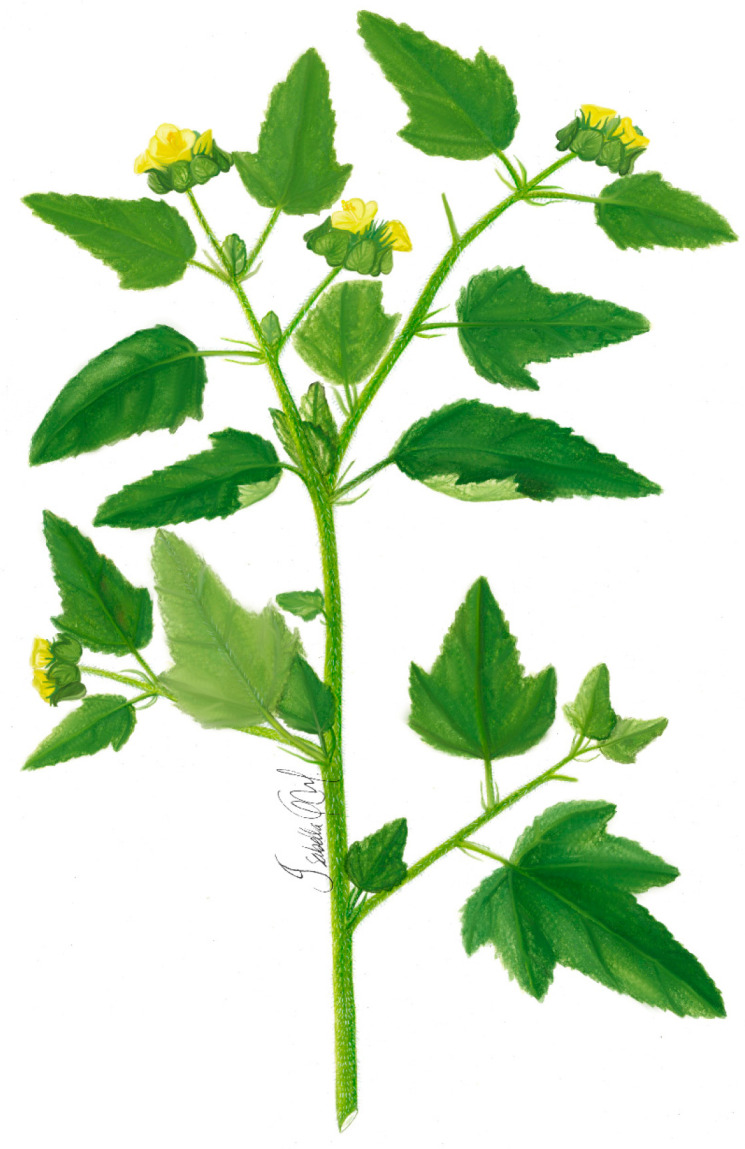
*Malachra rudis* Benth.

**Figure 7 plants-11-02808-f007:**
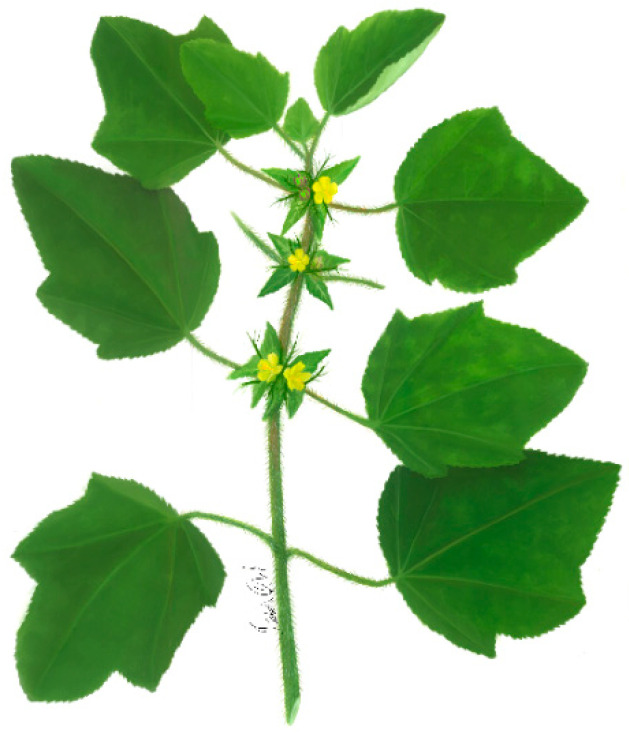
*Malachra urens* Poit.

**Figure 8 plants-11-02808-f008:**
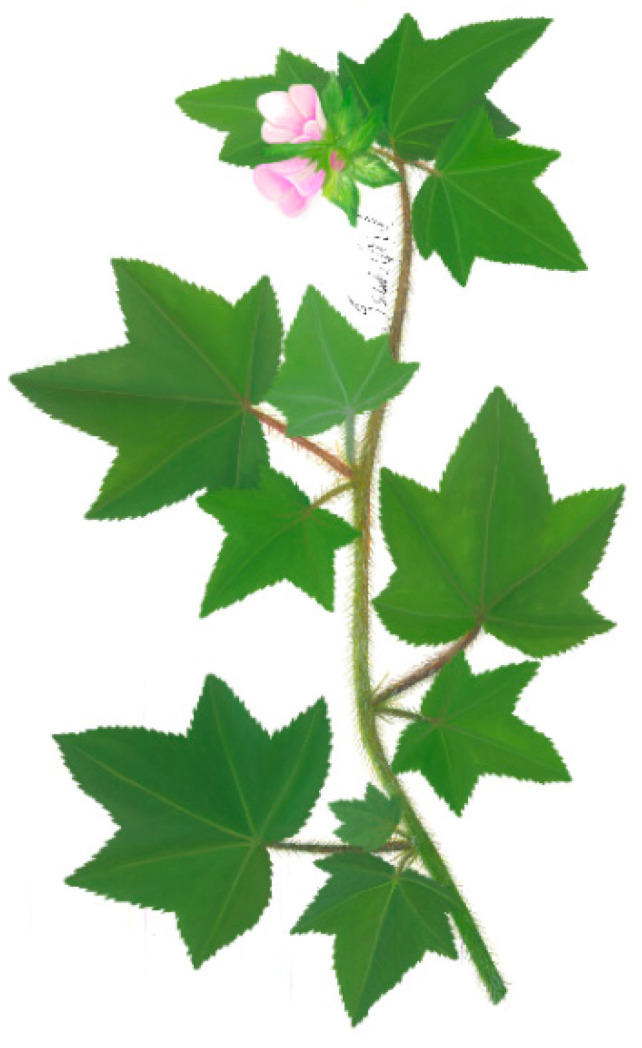
*Malachra helodes* Mart. ex Ledeb. & Alderstam.

**Figure 9 plants-11-02808-f009:**
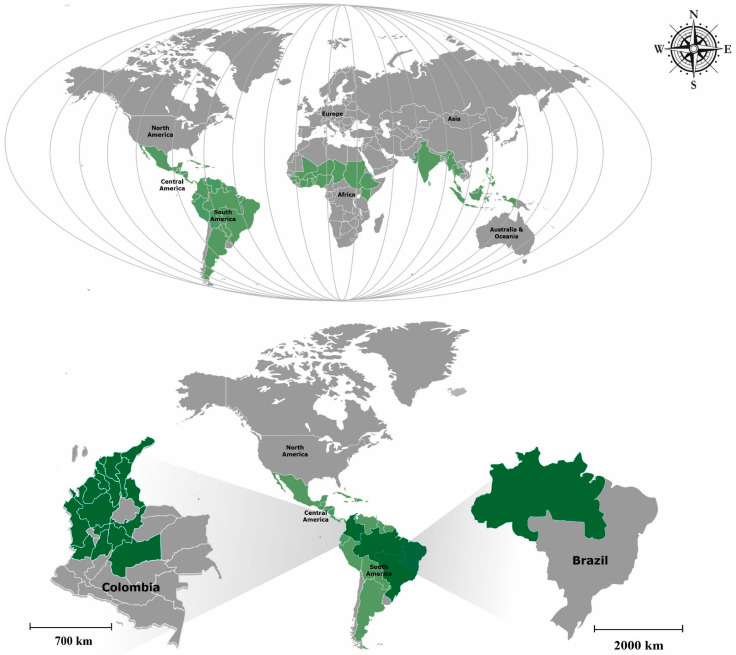
Geographical distribution of the genus *Malachra* L. Colombia and Brazil (green) countries with greater diversityThe countries of Colombia and Brazil have the largest number of species of the genus *Malachra* L., with six of the nine accepted names (*Malachra alceifolia* Jacq.; *Malachra capitata* (L.) L.; *Malachra fasciata* Jacq.; *Malachra radiata* (L.) L.; *Malachra ruderalis* Gürke Benth., with the different species *Malachra rudis* Benth. present only in Colombia [50] and the species *Malachra helodes* Mart. present only in Brazil [43]), followed by Peru, Ecuador, and Puerto Rico with five of the nine accepted names, see Table 3 [49,50].

**Figure 10 plants-11-02808-f010:**
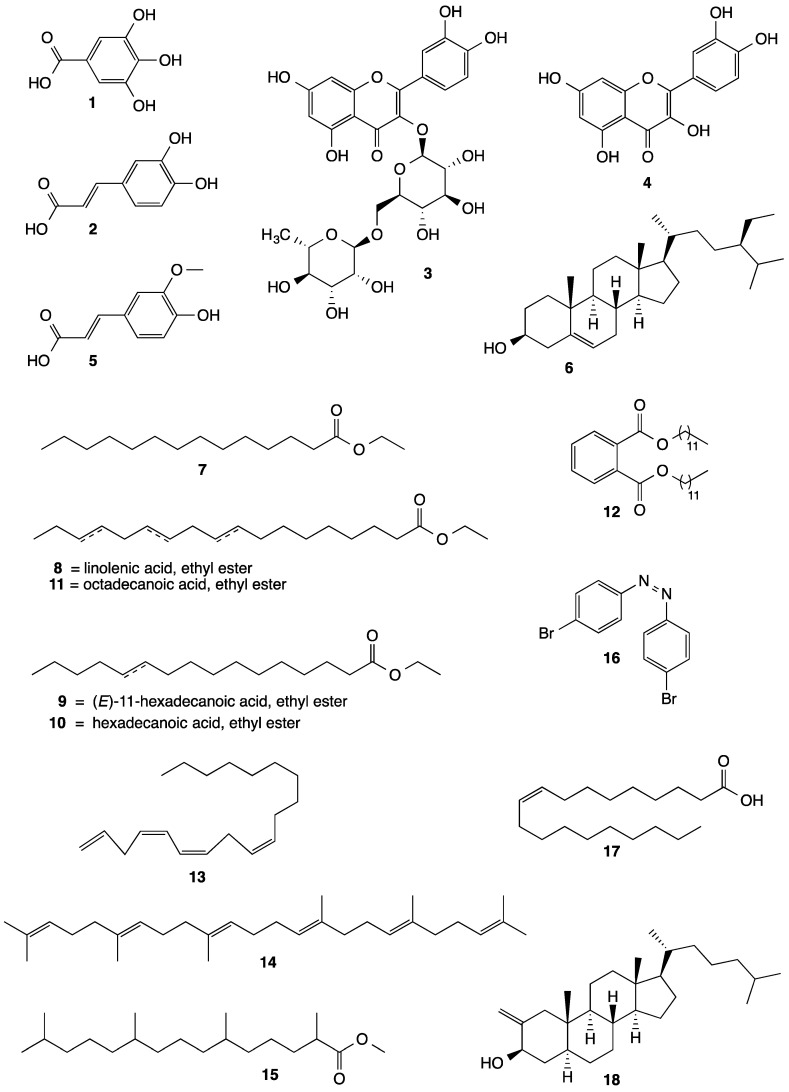
Molecular structure of some compounds isolated from species of the genus *Malachra* L.

**Table 1 plants-11-02808-t001:** Taxonomic name status of species in the genus *Malachra* L.

Genus	Species	Scientific Name Status		
		Accepted	Ambiguous	Unchecked
*Malachra* L.	*Malachra alceifolia* Jacq.			
	*Malachra capitata* L.			
	*Malachra digitata* C.Presl.			
	*Malachra fasciata* Jacq.			
	*Malachra helodes* Mart.			
	*Malachra lobata* L.			
	*Malachra officinalis* Klotzsch.			
	*Malachra plumosa* Desr.			
	*Malachra radiata* L.			
	*Malachra rosea* Hoffmans.			
	*Malachra ruderalis* Gürke.			
	*Malachra rudis* Benth.			
	*Malachra texana* A.Gray.			
	*Malachra triloba* [Desf.]			
	*Malachra urens* Poit. ex Ledeb. y Alderstam.			
	*Malachra urticifolia* C.Presl.			
	*Malachra viminea* Fisch. ex Steud.			

**Table 2 plants-11-02808-t002:** Taxonomic synonyms for species of the genus *Malachra* L.

Genus	Species	Taxonomic Synonyms
*Malachra* L.	*Malachra alceifolia* Jacq.	*Malachra alceifolia* var. *alceifolia* *Malachra alceifolia* var. *conglomerata* (Turcz.) Hochr. *Malachra alceifolia* var. *rotundifolia* (Schrank) Gürke. *Malachra conglomerata* Turcz. *Malachra hispida* Sessé & Moc. *Malachra rotundifolia* Schrank. *Urena capitata* var. *alceifolia* (Jacq.) M. Gómez.
	*Malachra capitata* L.	*Malachra heptaphylla* A. St.-Hil. *Malachra heptaphylla* Fisch. *Malachra mexicana* Schrad. *Malachra palmata* Moench. *Malachra velutina* Triana & Planch. *Sida capitata* L. *Urena capitata* (L.) M.Gómez. *Urena capitata* var. *capitata* *Urena moenchii* M.Gómez.
	*Malachra fasciata* Jacq.	*Malachra alceifolia* var. *fasciata* (Jacq.) A.Robyns. *Malachra fasciata* var. *lineariloba* (Turcz.) Gürke. *Malachra horrida* (Span.) Miq. *Malachra humilis* Benth. *Malachra kegeliana* Garcke. *Malachra lineariloba* Turcz. *Malva horrida* Span.
	*Malachra helodes* Mart.	No synonyms
	*Malachra radiata* L.	*Malachra bracteata* Cav. *Sida radiata* L. *Urena radiata* (L.) M. Gómez.
	*Malachra rudis* Benth.	*Malachra poeppigii* Gürke.
	*Malachra urens* Poit. ex Ledeb. y Alderstam.	*Urena urens* (Poit. ex Ledeb. & Alderstam) M. Gómez.

**Table 3 plants-11-02808-t003:** Global geographical distribution of the species of the genus *Malachra* L.

Species	Countries
*Malachra alceifolia* Jacq.	Belize, Brazil, Costa Rica, Colombia, Ecuador, Guyana, Guatemala, French Guiana, Ghana, Granada, India, Jamaica, Martinique, Mexico, Peru, Panama, Philippines, Puerto Rico, Suriname, Thailand and Venezuela,
*Malachra capitata* (L.) L.	Belize, Bolivia, Brazil, Costa Rica, Colombia, Ecuador, Guatemala, Guadalupe, Honduras, Mexico, Nicaragua, Peru, Panama, Puerto Rico and Thailand.
*Malachra fasciata* Jacq.	Belize, Bolivia, Brazil, Colombia, Costa Rica, Cuba, Dominican Re-public, Ecuador, El Salvador, French Guiana, Guatemala, Guyana, Haiti, Honduras, Jamaica, Leeward Isle, Mexico, Nicaragua, Pana-ma, Puerto Rico, Suriname, Trinidad-Tobago, Venezuela, Wind-ward Isle
*Malachra radiata* (L.) L.	Argentina, Benin, Bolivia, Brazil, Burkina, Chad, Colombia, Congo, Costa Rica, Cuba, Dominican Republic, El Salvador, Ghana, Guya-na, Haiti, Honduras, Ivory Coast, Jamaica, Mali, Mexico, Nicaragua, Niger, Nigeria, Panama, Paraguay, Peru, Puerto Rico, Senegal, Su-dan, Surinam, Togo, Venezuela
*Malachra ruderalis* Gürke.	Bolivia, Brazil, Colombia, Ecuador, Peru
*Malachra rudis* Benth.	Colombia, Ecuador, Peru.
*Malachra urens* Poit. ex Ledeb. & Alderstam.	Bahamas, Cuba, Dominican Republic, Florida, Haiti, Jamaica, Puerto Rico
*Malachra helodes* Mart.	Brazil

**Table 4 plants-11-02808-t004:** Species of the genus *Malachra* L. occurring in Colombia.

Scientific Name	Vernacular Name	Habit	Geographical Distribution	Ecology (IUCN) *	Conserv. Status (IUCN) **
*Malachra alceifolia* Jacq.	malauya, malva	Herb, subshrub	Native to Colombia, biogeographic region: Andean, Caribbean, Pacific 0–1800 m.a.s.l	Forest, woodland, savanna, shrubland, wetlands (inland), artificial–terrestrial.	LC (Least concern)
*Malachra capitata* (L.) L.	malva, malvavisco	Herb, subshrub	Native to Colombia, biogeographic region: Andean, Caribbean, Pacific, Magdalena valley 200–1480 m.a.s.l	Forest, woodland, shrubland, artificial–terrestrial.	Not evaluated
*Malachra fasciata* Jacq.	malva peluda, mano de muerto	Herb, subshrub	Native to Colombia, biogeographic region: Caribbean, Orinoquia, Magdalena valley 30–830 m.a.s.l	savanna, shrubland	Not evaluated
*Malachra radiata* (L.) L.	malva pata de gallina	Shrub	Native to Colombia, biogeographic region: Caribbean, Amazonia 0–200 m.a.s.l	Savanna	Not evaluated
*Malachra ruderalis* Gürke.	escoba, malva	Subshrub	Native to Colombia, biogeographic region: Andean, Pacific 5–1100 m.a.s.l	No report	Not evaluated
*Malachra rudis* Benth.	malva cimarrona, malva peluda, malvón	Subshrub	Native to Colombia, biogeographic region: Andean, Caribbean, Pacific, Magdalena valley, Amazonia 0–2330 m.a.s.l	Forest, woodland, savanna, shrubland, wetlands (inland), artificial–terrestrial	Not evaluated

* Habitat according to IUCN Habitats Classification (International Union Conservation of Nature). ** Conservation status according to IUCN Categories and Criteria (International Union Conservation of Nature).

**Table 5 plants-11-02808-t005:** Summary of the species of the genus *Malachra* L. with reported medicinal use.

*Malachra* Species	Vernacular Name	Part Used (Condition)	Principal Medicinal Indication/	Mode of Preparation	Way of Administration	Country	Reference
*Malachra alceifolia* Jacq.	Malva	Leaf	Inflammation and clogged nose	Decoction	Orally	Colombia	[1]
		Leaf and shoots	Malaria	Decoction	Orally	Peru	[31,32,33]
		Leaf	Childbirth, kidney disorder	Leaves minced with an egg-white (and soap) in water: bath	Bath	Perú	[34]
			Drunk headache, kidney disorder	leaves minced with *Citrus aurantifolia* fruit juice and water	Bath	Perú	[34]
			Headache, malaria, and fever	Leaves minced in water: bath	Bath	Perú	[34]
			Headache	leaves infused and applied as a poultice	Application locally	Perú	[34]
*Malachra capitata* (L.) L.	Malva of Horse, Malva-Xiu, Malachra, Yellow leaf bract, Brazil jute.	Roots, leaf	Pain, diarrhea, convulsion, hepatic cirrhosis, inflammation, pyrexia, ulcer, dementia, treatment of wounds, gastric disorders, jaundice embrocations for rheumatism lumbago, and febrifuge	Infusion, decoction	Orally	India, Perú	[53,54,55]
		Leaf	Childbirth, malaria, headache, and fever	Infusion, decoction	Orally	Perú	[34]
*Malachra fasciata* Jacq.	dead hand, hairy grass, malva Brava, wild okra.	Roots, leaf	Emollients, hemorrhoids, fever, and impotence	Infusion, decoction	Orally	Philippine’s	[29]
		Leaf	Gonorrhea, rheumatism, demulcent, and diuretic	Decoction	No report	No report	[30]
*Malachra ruderalis* Gürke.	Malva	Roots, fresh or dried	diseases of the respiratory tract: fever, cough, sore throat, flu, colds, pneumonia, whooping cough, pulmonary diseases, and COVID-19	Infusion, decoction	Orally	Perú	[35]
		Flowers	Stomach problems, diarrhea, fever, skin spots, skin infections, gastritis, and stress	No report	No report	Ecuador	[56]
		Whole plant	Inflammation and vaginal infection	Decoction	No report	Colombia	[57]

**Table 6 plants-11-02808-t006:** Major phytoconstituents identified in species of the genus *Malachra* L.

*Malachra* species	Plant Part Used	Phytoconstituents	Reference
*Malachra alceifolia* Jacq.	Leaf	Flavonoids, steroids, triterpenes anthocyanins, leucoanthocyanins, saponins	[22]
	Flowers	Flavonoids, steroids, triterpenes anthocyanins, leucoanthocyanins	
*Malachra capitata* (L.) L.	Root	Gallic acid **(1),** caffeic acid **(2),** rutin **(3),** quercetin **(4),** ferulic acid **(5)**	[23,24]
	Leaf	Rutin, ferulic acid	
	Stem	Gallic acid	
	Whole plant	*β*-Sitosterol **(6)**	[25]
	Root	Carbohydrates, phenols, flavonoids, glycosides, triterpenes, alkaloids, tannins, saponins	[23,24]
	Root	Tetradecanoic acid, ethyl ester **(7)**;	[23,24]
		linolenic acid, ethyl ester **(8)**;	
		(*E*)-11-hexadecenoic acid, ethyl ester **(9)**;	
		hexadecanoic acid, ethyl ester **(10)**;	
		octadecanoic acid, ethyl ester **(11)**;	
		didecyl phthalate **(12)**;	
		(*Z,Z,Z*)-1,4,6,9-nonadecatetraene **(13)**;	
		squalene **(14)**	
	Stem	Tetradecanoic acid, ethyl ester;	[27,28]
		pentadecanoic acid, 2,6,10,14- tetramethyl methyl ester **(15)**;	
		linolenic acid, methyl ester;	
		(*E*)-11-hexadecenoic acid, ethyl ester;	
		octadecanoic acid, ethyl ester;	
		(*Z,Z,Z*)-1,4,6,9-nonadecatetraene	
		azobenzene, 4,4′-dibromo- **(16)**;	
		squalene	
		oleic acid **(17)**	
		Cholestan-3-ol, 2-methylene-(3β,5α)- **(18)**;	
	Leaf	Tetradecanoic acid, ethyl ester	[27,28]
		3,7,11,15-tetramethyl-2-hexadecen-1-ol **(19)**;	
		Oxirane, tetradecyl- **(20)**;	
		(*E*)-11-hexadecenoic acid, ethyl ester	
		Hexadecanoic acid, ethyl ester	
		Phytol **(21);**	
		(*Z,Z*) 6,9 pentadecadien-1-ol **(22)**;	
		(*Z,Z*) 9,12-octadecadienoic acid **(23)**;	
		octadecanoic acid, ethyl ester;	
		squalene	
	Leaf	Flavonoids, glycosides, triterpenes, alkaloids, tannins, saponins, phlobatannins	[27,28]
*Malachra fasciata* Jacq.	Leaf	Aurantiamide acetate **(24)**	[29]
	Leaf	1,3-Diacylglycerol **(25)** 1,2-Diacylglycerol **(26)**	[30]

**Table 7 plants-11-02808-t007:** Summary of the species of the genus *Malachra* L. with reported pharmacological activity.

Malachra Species	Plant Part Used	Extract/Compounds	Pharmacological Activity	Concentration	Method	Major Findings	Reference
*M. alceifolia* Jacq.	Leaf	Ethanolic	Antiplasmodial	10 µg/mL	In vitro *Plasmodium falciparum* 152.2 ± 28.6 nM Chloroquine control	Inhibitory activity on *P. falciparum* ferriprotoporphyrin biomineralization inhibition	[31,32]
	Shoot	Ethanolic	Antiplasmodial	77 µg/mL	In vitro *Plasmodium falciparum* (3D7) chloroquine (concentration no report)	Inhibitory activity on *P. falciparum*	[33]
*M. capitata* (L.) L.	Leaf	Ethanolic	Antibacterial	62.5 ppm	In vitro MIC	Inhibition of the growth of *Propionibacterium acnes * (ATCC 6919)	[71]
	Shoot	Aqueous	Antidiarrhoeal	200 and 400 mg/kg	In vivo oral administration to Wistar rats; castor oil-induced diarrhoea, enteropooling, and small intestinal transit; 5 mL/kg, p.o diphenoxlate control	Decreases intestinal transit	[72]
		Aqueous	Anti-epileptic	250 and 500 mg/kg	In vivo maximal electroshock (MES) and pentylenetetrazole (PTZ)-induced seizuresmodels in albino Wistar rats, pentylenetetrazol control	Anticonvulsant activity against MES and PTZ animal models	[72]
		Aqueous	Anti-ulcerogenic	200 mg/kg and 400 mg/kg	In vivo oral administration to Wistar rats pylorus ligated model, 50mg/kg, p.o ranitidine control	Reduce the gastric acid secretion of pylorus	[72]
		Aqueous	Antioxidant	200 and 400 mg/kg	In vivo oral administration to Wistar rats	Inhibit the accumulation of lipid peroxidation product, superoxide dismutase, and catalase activities	[73]
		Aqueous	Hepatoprotective	100, 200 and 400 mg/kg	In vivo oral administration to Wistar rats, carbon tetrachloride CCl_4_ induced hepatotoxicity	Reduced levels of the hepatic enzymes SGOT, SGPT, alkaline phosphatase (ALP), and acid phosphatase (ACP)	[74]
	Leaf silver nanoparticles (AgNPs)	n-hexane	Antibacterial	1 mM	In vitro MIC	Nanoparticles bactericidal *Bacillus subtillis*, *Micrococcus Luteus, Staphylococcus aureus and Pseudomonas aeruginosa*	[75]
	Leaf	Methanolic, Chloroform Benzene	Antibacterial	50 mg/mL	In vitro MIC	Inhibition of the growth of *Escherichia coli* *Listeria monocytogenes*	[76]
*Malachra fasciata* Jacq.	Leaf	Chloroform/ Aurantiamide acetate	Antimicrobial	80 μg	In vitro MIC	Inhibition of the growth of *P. aeruginosa, B. subtilis, C. albicans*,	[29]
	Leaf	Unknown	Phototoxic (photosensitization)	Unknown	In vivo sheep females and male	Cause of primary photodermatitis in sheep ingestion leaf	[77]
	Leaf	Chloroform	Antifungal	30 µg	In vitro MIC	Inhibition of the growth of *Aspergillus niger*	[29]
	Leaf	-(-)loliolide	Antimutagenic	8 mg/kg	In vitro micronucleus test induced using mitomycin C, 32%	Reduce the number of micronucleated polychromatic erythrocytes (MPCE)	[30]

## Data Availability

Not applicable.

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
