# Peer review of "An Overview of Genus *Malachra* L.—Ethnobotany, Phytochemistry, and Pharmacological Activity"

_plants, 2022, doi:10.3390/plants11212808_

Round 1

Reviewer 1 Report

The manuscript "An overview of genus Malachra: Ethnobotany, Phytochemistry and Pharmacological activity" is an interesting review that aimed at providing new insight into the ethnomedicine, photochemistry, and pharmacology of the genus Malachra. The manuscript presents the systematic taxonomy and distribution of Malachra species and emphasizes the potential of the extracts from presented Malachra species as nutraceuticals used in medicinal use.   However, the innovation and depth are very low. Although the manuscript shows interesting points of view, there are areas which still need a lot of improvements. It must be acknowledged that they did a lot of work on the data summary. It is not little but not enough. Some points are interesting,  but remember that a review must be a critical analysis of the data and do not simply provide a long list of (certainly interesting) facts.

Major revision

1.  Systematic reviews should follow PRISMA guideline as indicated in the instruction for authors. Concerning the ethnobotany, phytochemistry and pharmacological activity of six out of the eight Malachra species in Colombia, do you find data only about 6 species, Malachra alceifolia Jacq; Malachra capitata (L.) L., Malachra fasciata Jacq., Malachra radiata (L.) L., Malachra ruderalis Gürke., and Malachra rudis Benth. It was complete data mining concerning phytochemistry or ethnomedicine? No data is available regarding extraction of articles, eligible criteria, selection methodology, rejected papers, etc. See above PRISMA.

2.  The chemical structure of major Malachra species phytoconstituents  should be drawn using professional drawing software, like ChemDraw.

3.  Malachra species with pharmacological activity- This chapter is an interesting list of information, but without a discussion able to point up some significant findings. Like the ethanol extract of Malachra alceifolia Jacq leaf possess antibacterial activity. What kind of bacterial can the extract inhibit, how to inhibit, what is the inhibitory dose, method of experiment and positive and negative control.

Table 5 should be improved by adding data concerning dose.

4.  The manuscript even if well done, but it must be understood that there must be an element of scientific novelty (new research trends) as an intellectual effort and a precise methodology to conduct the literature digging. In my opinion, there is no scientific news worthy of publication if the work does not hypothesize new lines of research, these data are a summary.

Minor revision

Line 24 -potencially-potentially

Please modify the all references in the main text and make sure the ref appear in the numberial order. Such as ref 8 is after ref 16 (line 43-44)

Line 40- containin-containing

Line 45- “plant”-“plants

Line 63-72 - For a separate paragraph describing the methodology, it would be better to add a figure describing the selective process. Like "How much literature was collected", "The inclusion criteria of screening and selecting eligible articles"

Line 80- Delete of

Author Response

Response to Reviewer 1 Comments

Point 1: Systematic reviews should follow PRISMA guideline as indicated in the instruction for authors. Concerning the ethnobotany, phytochemistry and pharmacological activity of six out of the eight Malachra species in Colombia, do you find data only about 6 species, Malachra alceifolia JacqMalachra capitata (L.) L.Malachra fasciata Jacq., Malachra radiata (L.) L., Malachra ruderalis Gürke., and Malachra rudis Benth. It was complete data mining concerning phytochemistry or ethnomedicine? No data is available regarding extraction of articles, eligible criteria, selection methodology, rejected papers, etc. See above

Response 1: PRISMA methodology is attached for Point 1

Point 2: The chemical structure of major Malachra species phytoconstituents  should be drawn using professional drawing software, like ChemDraw. 

Response 2: The change has been made for Point 2

Point 3: Malachra species with pharmacological activity- This chapter is an interesting list of information, but without a discussion able to point up some significant findings. Like the ethanol extract of Malachra alceifolia Jacq leaf possess antibacterial activity. What kind of bacterial can the extract inhibit, how to inhibit, what is the inhibitory dose, method of experiment and positive and negative control.

Response 3: Table 7.  Summary of Malachra species with pharmacological activity reported.for Point 3.

Point 4: Table 5 should be improved by adding data concerning dose.

Response 4: Table 7.  Summary of Malachra species with pharmacological activity reported

Point 5: The manuscript even if well done, but it must be understood that there must be an element of scientific novelty (new research trends) as an intellectual effort and a precise methodology to conduct the literature digging. In my opinion, there is no scientific news worthy of publication if the work does not hypothesize new lines of research, these data are a summary.

Response 5: Summary adjusted

 Point 6: Line 24 -“potencially”-“potentially”

Response 6: Summary adjusted

Point 7: Please modify the all references in the main text and make sure the ref appear in the numberial order. Such as ref 8 is after ref 16 (line 43-44)

Response 7: References corrected

Point 8: Line 40- “containin”-“containing”

Response 8: Line corrected

Point 9: Line 45- “plant”-“plants”

Response 9: Line corrected

Point 10: Line 63-72 - For a separate paragraph describing the methodology, it would be better to add a figure describing the selective process. Like "How much literature was collected", "The inclusion criteria of screening and selecting eligible articles"

Response 10: Line corrected

Point 11: Line 80- Delete “of”

Response 11: Line corrected

Reviewer 2 Report

Interesting review of the Malachara species and literature.  My biggest concern is the grammar/ small spelling errors in the article (ie. in the abstract, potentially is written as potencially and that sentence may be better written as "The species...  potentially have many biological activities..."

ie. Grammer issue:  "The populations of developing countries" should be fixed, this sentence does not sound right.

ie. puncutation issue:  that second line, there is not space between the citation "[2,62].Products"

Author Response

Response to Reviewer 2 Comments

Point 1: Comments and Suggestions for Authors Interesting review of the Malachara species and literature.  My biggest concern is the grammar/ small spelling errors in the article (ie. in the abstract, potentially is written as potencially and that sentence may be better written as "The species...  potentially have many biological activities..."

  1. Grammer issue:  "The populations of developing countries" should be fixed, this sentence does not sound right.

Response 1: Line corrected

Point 2: ie. puncutation issue:  that second line, there is not space between the citation "[2,62].Products

Response 2: Line corrected

Reviewer 3 Report

Submitted review deals with the topic of ethnobotany and main secondary metabolites identified in genus Malachra and their biological activities. Although the topic is interesting and not much researched as the plants from genus Malachra have not yet been studied very intensively, major revisions of language, style, completeness of information and conclusions are needed, if the review is to be published.

At the beginning I would like to mention some overall notes:      

1. At first, it is necessary to rearrange references order in text. In the introduction, there are references 1, 2 and followed by references number 62, 7 and 50.

2. I recommend to put references through the paragraphs regarding the key information, not 7 references at the end of the paragraph.

3. I recommend paying extra attention to the language proofreading of the text. Some sentences are very long with only few verbs which causes a poor understanding of the text or sometimes a loss of meaning.

4. There is italics missing sometimes in th genus name etc.

5. Figures and formulas are blurred – it is necessary to improve the resolution.

6.  If there should be cross-links to the figures, there should be all active – they are not in fig. 6, 7 and 9.

7. I recommend to go through all tables, unify the font size and adjust column width or if necessary, I suggest turning some tables by 90 degrees to get enough space for all the necessary information.

8. All formulas look like they were downloaded from various sources. It is necessary to draw them all in uniform style and format.

Comments and questions:

Line 38: Authors mentioned that „80% of the world's population uses medicinal plants as the only source of treatment for various diseases“ but the reference speaks about 80% of population in developing countries not worldwide.

Line 40: I recommend to specify in more detail, why authors decided to mention 8 accepted species from genus Malachra, when worlfloraonline.org which they cite [ref 56] mentions 17 species in the genus. Authors also mention 43 species in the genus – where is this information from?

Line 45: What authors mean by this sentence: „The location of chemical compounds present in Malachra L. varies among different plant structures…“?

Lines 50-54: There is mixture of states and continents in this pararaph – unify it.

Lines 86-149: There is not a single verb in these few paragraphs! Authors should put these information to the table or use shorter sentences with verbs.

Lines 90, 102: I would rather use word venation instead of nerves (significant venation, palmate venation, etc.).

Lines 94,116, 123, 131, 150, 156, 162 and 169: I recommend to indent the paragraphs/text following the names of plant species.

Line 117: Authors should use a multiplication sign instead of a letter „X“.

Lines 204-214: Is this paragraph necessary? Why there are 50 mentioned countries when we can see them in map (Fig 10)?

Line 221: Table header is missing.

Lines 223-226: This information is already mentioned (lines 60-62) – this is unnecessary duplication.

Line 235: Already mentioned (with the mistake) in lines 38-39.

Line 357: Tab. 5 In this part I would recommend to include some numerous values, if they are available, as from this table the reader is not able to find out, how significant tested activities of Malachra spp. are (compared to proper standards). I would also discuss these activities/values in conclusions (lines 373-375).

Author Response

Response to Reviewer 3 Comments

Submitted review deals with the topic of ethnobotany and main secondary metabolites identified in genus Malachra and their biological activities. Although the topic is interesting and not much researched as the plants from genus Malachra have not yet been studied very intensively, major revisions of language, style, completeness of information and conclusions are needed, if the review is to be published.

At the beginning I would like to mention some overall notes:     

Point 1: At first, it is necessary to rearrange references order in text. In the introduction, there are references 1, 2 and followed by references number 62, 7 and 50.

Response 1: corrected

Point 2: I recommend to put references through the paragraphs regarding the key information, not 7 references at the end of the paragraph.

Response 2: corrected

Point 3: I recommend paying extra attention to the language proofreading of the text. Some sentences are very long with only few verbs which causes a poor understanding of the text or sometimes a loss of meaning.

Response 3: corrected

Point 4: There is italics missing sometimes in th genus name etc.

Response 4: corrected

Point 5: Figures and formulas are blurred – it is necessary to improve the resolution.

Response 5: figures replaced by table 1 and 2, corrected

Point 6: If there should be cross-links to the figures, there should be all active – they are not in fig. 6, 7 and 9.

Response 6: improved

Point 7: I recommend to go through all tables, unify the font size and adjust column width or if necessary, I suggest turning some tables by 90 degrees to get enough space for all the necessary information.

Response 7: improved

Point 8: All formulas look like they were downloaded from various sources. It is necessary to draw them all in uniform style and format.

 Response 8: improved

Point 9 : Line 38: Authors mentioned that „80% of the world's population uses medicinal plants as the only source of treatment for various diseases“ but the reference speaks about 80% of population in developing countries not worldwide.

Response 9: paragraph adjusted and improved

Point 10 : Line 40: I recommend to specify in more detail, why authors decided to mention 8 accepted species from genus Malachra, when worlfloraonline.org which they cite [ref 56] mentions 17 species in the genus. Authors also mention 43 species in the genus – where is this information from?

Response 10: paragraph adjusted and improved

Point 11 : Line 45: What authors mean by this sentence: „The location of chemical compounds present in Malachra L. varies among different plant structures…“?

Response 11: paragraph adjusted and improved

Point 12 : Lines 50-54: There is mixture of states and continents in this pararaph – unify it. Response 12:  paragraph adjusted and improved

Point 13 : Lines 86-149: There is not a single verb in these few paragraphs! Authors should put these information to the table or use shorter sentences with verbs.

Response 13:  paragraph adjusted and improved

Point 14 : Lines 90, 102: I would rather use word venation instead of nerves (significant venation, palmate venation, etc.).

Response 14:  paragraph adjusted and improved

Point 15 : Lines 94,116, 123, 131, 150, 156, 162 and 169: I recommend to indent the paragraphs/text following the names of plant species.

Response 15:  paragraph adjusted and improved

Point 16 : Line 117: Authors should use a multiplication sign instead of a letter „X“.

Lines 204-214: Is this paragraph necessary? Why there are 50 mentioned countries when we can see them in map (Fig 10)?

Response 16:  corrected

Point 17 : Line 221: Table header is missing.

Response 17:  corrected

Point 18 : Lines 223-226: This information is already mentioned (lines 60-62) – this is unnecessary duplication.

Response 18:  corrected

Point 19 : Line 235: Already mentioned (with the mistake) in lines 38-39.

Response 19:  corrected

Point 20 : Line 357: Tab. 5 In this part I would recommend to include some numerous values, if they are available, as from this table the reader is not able to find out, how significant tested activities of Malachra spp. are (compared to proper standards). I would also discuss these activities/values in conclusions (lines 373-375).

Response 20:  Table 7.  Summary of Malachra species with pharmacological activity reported.

Reviewer 4 Report

This review paper is very interesting because it is a good contribution for the knowledge about the taxonomy, botanical descriptions, geographical distribution, phytochemistry, ethnobotany and pharmacological activities of Malachra species.

The article has ten Figures, five Tables and presents an interesting study well discussed and which summarized a huge amount of information from important electronic databases and other sources. The conclusion was supported by the bibliographic references.

There are some minor changes that need to be made but which are of the presentation nature. The proposed review changes are as follow:

- On page 1 (line 42), page 2 (line 62), page 4 (line150), page 8 (line 225) “Gürke.” should be “Gürke”;

- On page 2 (line 61) “Jacq” should be “Jacq.”; (line 65) “Malachra” should be in italic "Malachra";

- On Figure 6 (page 6) “Gürke.” should be “Gürke”;

- On Figure 7 (page 6) “Benth” should be “Benth.”;

- On Table 1 (page 8), first column, “Jacq” should be “Jacq.”, “L” should be “L.”, “Benth” should be “Benth.”, and “Mart” should be “Mart.”;

- On Table 2 (page 9), first column, “Gürke.” should be “Gürke”;

- On Table 3 (pages 10 and 11), first column, “Jacq” should be “Jacq.” and “L” should be “L.”; in sixth column, “Perú” should be “Peru”; fifth column, “Citrus aurantifolia” should be in italic "Citrus aurantifolia";

- On Table 5 (page 17), first column, “Gürke.” should be “Gürke” and third column, “chloroformo” should be “chloroform”;

- On page 18, line 361, “Jac” should be “Jacq.”, line 363 “Gürke.” should be “Gürke”, lines 369 and 370, “Malachra L” should be “Malachra L.”.

With these minor changes, the recommendation will be to accept the manuscript for publication.

Author Response

Response to Reviewer 4 Comments

Point 1. On page 1 (line 42), page 2 (line 62), page 4 (line150), page 8 (line 225) “Gürke.” should be “Gürke”;

Response 1: Line corrected

Point 2. On page 2 (line 61) “Jacq” should be “Jacq.”; (line 65) “Malachra” should be in italic "Malachra";

Response 2: Line corrected

Point 3. On Figure 6 (page 6) “Gürke.” should be “Gürke”;

Response 3: Line corrected

Point 4. On Figure 7 (page 6) “Benth” should be “Benth.”;

Response 4: Line corrected

Point 5. On Table 1 (page 8), first column, “Jacq” should be “Jacq.”, “L” should be “L.”, “Benth” should be “Benth.”, and “Mart” should be “Mart.”;

Response 5: Line corrected

Point 6. On Table 2 (page 9), first column, “Gürke.” should be “Gürke”;

Response 6: Line corrected

Point 7. On Table 3 (pages 10 and 11), first column, “Jacq” should be “Jacq.” and “L” should be “L.”; in sixth column, “Perú” should be “Peru”; fifth column, “Citrus aurantifolia” should be in italic "Citrus aurantifolia";

Response 7: Line corrected

Point 8.  On Table 5 (page 17), first column, “Gürke.” should be “Gürke” and third column, “chloroformo” should be “chloroform”;

Response 8: Line corrected

Point 9.  On page 18, line 361, “Jac” should be “Jacq.”, line 363 “Gürke.” should be “Gürke”, lines 369 and 370, “Malachra L” should be “Malachra L.”.

Response 9: Line corrected

Round 2

Reviewer 1 Report

The revised manuscript "An overview of genus Malachra: Ethnobotany, Phytochemistry and Pharmacological activity" provides detailed information about the ethnobotanical, phytochemical, biological, pharmacological, and toxicological information of the genus Malachra, offering the pharmacognostic potential for the search of new drugs and the development of innovative medicines. The authors used a significant number of proper references, and added a separate paragraph for describing the methodology and the selective process to make the manuscript dramatically improved in terms of hierarchy and structure. The botanical description and geographical distribution of Malachra species are detailed and illustrated with a comprehensive and clear chart. The manuscript also introduced the ethnobotanical (medicinal) use, pharmacological activity, phytochemical in detail with some comprehensive tables. Conclusions are clear and consistent. The English level, the novelty, and the standards of the work are enough to be published.The revised manuscript "An overview of genus Malachra: Ethnobotany, Phytochemistry and Pharmacological activity" provides detailed information about the ethnobotanical, phytochemical, biological, pharmacological, and toxicological information of the genus Malachra, offering the pharmacognostic potential for the search of new drugs and the development of innovative medicines. The authors used a significant number of proper references, and added a separate paragraph for describing the methodology and the selective process to make the manuscript dramatically improved in terms of hierarchy and structure. The botanical description and geographical distribution of Malachra species are detailed and illustrated with a comprehensive and clear chart. The manuscript also introduced the ethnobotanical (medicinal) use, pharmacological activity, phytochemical in detail with some comprehensive tables. Conclusions are clear and consistent. The English level, the novelty, and the standards of the work are enough to be published.

Author Response

Color changes in the wording of the botany, geographical distribution and map are checked to improve the document.

Reviewer 3 Report

This version of the manuscript is much better than the previous one but there are still some things to improve.

In several cases authors just did not take into account my previous comments and suggestions without explaining but with answer „corrected“. I am not going to repeat all of my previous comments but I would like to mention some of them AGAIN:

1. I recommend another english check as there are still many missing verbs and prepositions in the text.

2.There is italics missing sometimes in the genus names etc.

3. In part 4. Botanical description of Malachra species there are still almost any verbs used and I find it not very well-arranged and confusing for the reader.

Additional comments:

Line 28:  „… animal animals“

Line 37-39: This sentence is not clear.

Lines 97-99, 102-105, etc.: Why do you use quotation marks here? It shoul not be used in taxonomy. (Malachra alceifolia "Jacq.", etc.).

Lines 272-275: Why there are so many brackets in this sentence?

Lines 288-289: What i is meant by this sentence?

Line 459: Align figures description with the rest of the text.

Tables: When table continues from one page to another, there should be something like Tab. 6 cont. and the table header should be repeated on the subsequent page.

Author Response

2.There is italics missing sometimes in the genus names etc.

Response 2: Corrected the line of gender names, made color changes in wording

  1. In part 4. Botanical description of Malachraspecies there are still almost any verbs used and I find it not very well-arranged and confusing for the reader.

Response 3: Corrected the line of Botanical description of Malachra species, made color changes in wording

Additional comments:

Line 28:  „… animal animals“

Response 3: Corrected the line of animals in the abstract, made color changes in wording

Line 37-39: This sentence is not clear.

Response 4: Corrected the line sentence, made color changes in wording

Lines 97-99, 102-105, etc.: Why do you use quotation marks here? It shoul not be used in taxonomy. (Malachra alceifolia "Jacq.", etc.).

Response 5: Corrected line phrasing, made color changes to wording, removed "Jacq.",

Lines 272-275: Why there are so many brackets in this sentence?

Response 6: Corrected the line sentence, made color changes in wording

Lines 288-289: What i is meant by this sentence?

Response 7: Corrected the line sentence, made color changes in wording

Line 459: Align figures description with the rest of the text.

Response 8: Corrected the line align figures 1 -8 description

Tables: When table continues from one page to another, there should be something like Tab. 6 cont. and the table header should be repeated on the subsequent page.

Response 7: Corrected the line table 6, made color changes in wording
